# Reduced *SOCS1* Expression in Lung Fibroblasts from Patients with IPF Is Not Mediated by Promoter Methylation or Mir155

**DOI:** 10.3390/biomedicines9050498

**Published:** 2021-04-30

**Authors:** Cecilia M. Prêle, Thomas Iosifidis, Robin J. McAnulty, David R. Pearce, Bahareh Badrian, Tylah Miles, Sarra E. Jamieson, Matthias Ernst, Philip J. Thompson, Geoffrey J. Laurent, Darryl A. Knight, Steven E. Mutsaers

**Affiliations:** 1Institute for Respiratory Health, Nedland, WA 6009, Australia; cecilia.prele@uwa.edu.au (C.M.P.); thomas.iosifidis@uwa.edu.au (T.I.); bahareh.badrian@nd.edu.au (B.B.); tylah.miles@research.uwa.edu.au (T.M.); philip.thompson@lunghealth.net.au (P.J.T.); geoff.laurent@uwa.edu.au (G.J.L.); 2Centre for Respiratory Health and Centre for Cell Therapy and Regenerative Medicine, School of Biomedical Sciences, University of Western Australia, Nedland, WA 6009, Australia; 3Centre for Inflammation and Tissue Repair, Rayne Institute, Department of Medicine, University College London, London WC1E 6JJ, UK; r.mcanulty@ucl.ac.uk (R.J.M.); david.pearce@ucl.ac.uk (D.R.P.); 4Telethon Kids Institute and Centre for Child Health Research, University of Western Australia, Nedlands, WA 6009, Australia; sarra.jamieson@uwa.edu.au; 5Olivia Newton-John Cancer Research Institute, Heidelberg, VIC 3084, Australia; Matthias.Ernst@onjcri.org.au; 6Faculty of Medicine, University of British Columbia (UBC), Vancouver, BC V6Z 1Y5, Canada; Darryl.Knight@hli.ubc.ca

**Keywords:** L-6, Jak/STAT pathway, SOCS1, miR155, fibroblast, fibrosis

## Abstract

The interleukin (IL)-6 family of cytokines and exaggerated signal transducer and activator of transcription (STAT)3 signaling is implicated in idiopathic pulmonary fibrosis (IPF) pathogenesis, but the mechanisms regulating STAT3 expression and function are unknown. Suppressor of cytokine signaling (SOCS)1 and SOCS3 block STAT3, and low SOCS1 levels have been reported in IPF fibroblasts and shown to facilitate collagen production. Fibroblasts and lung tissue from IPF patients and controls were used to examine the mechanisms underlying SOCS1 down-regulation in IPF. A significant reduction in basal *SOCS1* mRNA in IPF fibroblasts was confirmed. However, there was no difference in the kinetics of activation, and methylation of SOCS1 in control and IPF lung fibroblasts was low and unaffected by 5′-aza-2′-deoxycytidine’ treatment. *SOCS1* is a target of microRNA-155 and although microRNA-155 levels were increased in IPF tissue, they were reduced in IPF fibroblasts. Therefore, SOCS1 is not regulated by *SOCS1* gene methylation or microRNA155 in these cells. In conclusion, we confirmed that IPF fibroblasts had lower levels of SOCS1 mRNA compared with control fibroblasts, but we were unable to determine the mechanism. Furthermore, although SOCS1 may be important in the fibrotic process, we were unable to find a significant role for SOCS1 in regulating fibroblast function.

## 1. Introduction

Idiopathic pulmonary fibrosis (IPF) is a progressive and fatal disease, and despite an increasing number of clinical trials over the past 15 years, has limited treatment options [1,2]. IPF belongs to a group of heterogeneous lung diseases characterized by an excessive deposition of extracellular matrix (ECM) proteins within the pulmonary interstitium, leading to impaired gas exchange and loss of lung function. The pathogenesis of IPF remains poorly understood but disease progression is closely associated with regions of fibroblast accumulation and proliferation [1,3]. We have previously reported that Janus kinases (Jak)/signal transducer and activator of transcription (STAT)-mediated signaling, following the binding of the IL-6 family of cytokines to specific cell surface receptors and the shared signal transducing subunit gp 130, drives lung fibrosis in vivo [4] and that a population of fibroblasts from distinct regions of the fibrotic lung are characterized by heightened STAT3 expression [5]. 

The Jak/STAT pathway is regulated by three main families of proteins: the suppressor of cytokine signaling (SOCS) family, the protein inhibitors of activated signaling (PIAS), and SH2 domain containing phosphatases. These endogenous inhibitor molecules act to negatively regulate the self-perpetuating response induced by STAT transcriptional activity and act at different steps of the activated pathway [6]. 

The SOCS family consist of eight structurally related proteins that are rapidly synthesized following STAT-mediated gene transcription [7] and act to inhibit STAT phosphorylation and signal transduction by at least two distinct mechanisms—interaction with the catalytic domains of JAK proteins to inhibit their activity, thereby preventing the subsequent activation of STAT3 [8], or by binding to gp130 and elongins, which form part of the E3 ubiquitin-ligase complex that target ubiquitinated proteins for degradation [9]. SOCS proteins are highly specific, with different SOCS molecules controlling the activity of specific cytokines. In myeloid cells, responses to interferon (IFN)-γ and interleukin (IL)-4 are subject to the negative regulatory effects of SOCS1, while responses to IL-6 and granulocyte colony stimulating factor (G-CSF) are controlled by SOCS3 [10,11]. SOCS1 binds to the Jak tyrosine kinase domain and inhibits catalytic activity via a kinase inhibitory region adjacent to the SH2 domain [12], which itself is a transcriptional target for STAT and provides a negative feedback loop. Initially the regulatory effects of SOCS1 and SOCS3 proteins on STAT activity were thought to be specific, with SOCS3 negatively regulating STAT3 activity and SOCS1 controlling STAT1 activity. More recently, a role for SOCS1 in the negative regulation of STAT3 signal transduction was also described [13]. SOCS1-expression is also reduced in a number of cancers as a result of promoter methylation [13,14,15]. For example, silencing of SOCS1 is associated with altered STAT3 activation in the development of hepatocellular carcinoma (HCC) [13]. Furthermore, the SOCS1 peptide mimetic Tkip has been shown to inhibit IL-6-induced activation of STAT3 in prostate cancer cells by binding to the autophosphorylation site of JAK2 [16].

A study using haplodeficient *socs1* mutant mice has implicated this protein in the development of lung fibrosis [17]. This is supported by the finding that SOCS1 expression is reduced in peripheral blood mononuclear cells from IPF patients [18] and in IPF lung biopsy tissue and fibroblasts isolated from IPF patients [17]. Furthermore, reduced *socs1* has been associated with increased collagen production by mouse lung fibroblasts [19]. However, the mechanisms underlying the reduced expression of SOCS1 is not known.

In this study, we used primary fibroblast cultures from normal and IPF lungs to investigate several mechanisms that are likely to contribute to reduced SOCS1 expression in IPF fibroblasts, including the kinetics of *SOCS1* induction, *SOCS1* promoter methylation, and the role of microRNA (miR)-155 as a negative regulator of *SOCS1*. We confirmed that *SOCS1* mRNA levels are reduced in fibroblasts established from IPF patients compared to control fibroblasts, but this was not due to changes in the kinetics of induction of *SOCS1* mRNA, altered *SOCS1* promoter methylation, or regulation by miR-155. Therefore, the mechanism regulating SOCS1 downregulation in these cells was not determined. We also examined the effect of SOCS1 on collagen production in the fibroblast cell lines, but in contrast to previous reports [19], reduced *SOCS1* levels did not correlate with increased collagen expression by IPF fibroblasts, questioning the role of SOCS1 in these cells.

## 2. Methods

### 2.1. Cell Cultures and Lung Tissue Samples

Primary lung fibroblast cultures (6 IPF and 9 control) were either purchased from American Type Culture Collection (ATCC, Manassas, VA, USA)—LL-86 (CCL-190), Hs888Lu (CRL-7624), CCD-13Lu (CCL-200), CCD-16Lu (CCL-204), CCD-19Lu (CCL-210), LL-29 (CCL-134), LL-97A (CCL-191)—or provided by Professor Robin McAnulty (University College London, London, UK), and had been isolated and cultured using the protocol as previously described [20]. IPF fibroblasts were isolated from fibrotic lung tissue from patients undergoing surgical lung biopsy or transplant surgery (aged 62 ± 4 years). Control cells were isolated from lung tissue obtained from histologically normal areas of peripheral lung removed at lung cancer resection (aged 59 ± 7 years). All tissue was obtained with appropriate informed consent and its use approved by the East Midlands, Nottingham 2 NRES Committee, Ref. 12/EM/0058, 26 January 2012. All experiments were conducted in accordance with the terms of the informed consents and with relevant guidelines and regulations. miR-155 transfection studies were performed on ATCC CCD-19Lu fibroblasts.

Cells were cultured on standard tissue culture plastic and maintained in DMEM (Invitrogen Life Technologies, Mulgrave, VIC, Australia) supplemented with 10% fetal calf serum (FCS), 4 mM of L-glutamine, and 5 μg/mL of gentamicin (Sigma-Aldrich, Castle Hill, NSW, Australia) and used up to passage 10. For RNA and protein assays, cells were grown to confluence in 6-well tissue culture plates, the medium changed to DMEM containing 0.4% FCS and standard supplements overnight and cells were treated, as outlined in the figure legends.

The molecular crowding assay (scar-in-a-jar assay) of lung collagen deposition was performed as previously described [21]. Collagen deposition was detected by staining for collagen 1 (Sigma, C2456) and visualized with Alexa Fluor 488 (Thermo Scientific, Rockland, IL, USA; A10011). The cells were counterstained with DAPI (Invitrogen; 1D306) and normalized to total cell count. The images were quantified using Image Xpress (Molecular Devices, San Jose, CA, USA). Formalin-fixed, paraffin-embedded (FFPE) lung tissue sections from patients with a confirmed diagnosis of usual interstitial pneumonia or control resected lung were obtained from retrospective diagnostic biopsy specimens from PathWest Laboratory Medicine (Perth, WA, Australia). Ethics approval Bellberry Limited Human Research Ethics Committee, ref 2011-10-497, 01 February 2012. Patient consent for tissues sections used in this study was not required, in accordance with National Health and Medical Research Committee (NHMRC) guidelines.

### 2.2. Isolation of RNA and DNA from FFPE Tissue Samples

FFPE tissues sections (15 μm) were de-paraffinized and RNA harvested using the High Pure FFPE RNA micro kit (Roche Diagnostics, North Ryde, NSW, Australia). 

### 2.3. Quantitative Real Time PCR Analysis

RNA was prepared using Qiagen RNAeasy isolation columns and reverse transcribed to cDNA using Omniscript reverse transcriptase (Qiagen, Germantown, MD, USA) and random hexamer primers (Invitrogen). Real time PCR was performed on a Step one plus real time PCR machine (Applied Biosystems, Waltham, MA, USA) using Taqman probes. For miRNA analysis, cells were lysed in QiaZol reagent (Qiagen) and miRNA extracted using the Qiagen miRNeasy Mini Kit and as previously described [22]. miR155 expression was determined using miRNA PCR mix and FAM miR155-specific primers (Invitrogen; 002623).

### 2.4. Western Blot Analysis of Activated STAT Expression

Cells were grown to confluence and quiesced overnight prior to stimulation with 10 ng/mL IL-6 (Peprotech, Rehovot, Israel) or 20 ng/mL IFNγ (Sigma-Aldrich), which activate STAT3 and STAT1, respectively. The cells were lysed with buffer containing 10 mM Tris base, 50 mM NaCl, 5 mM EDTA, 1% Triton X-100 supplemented with 1 mM sodium pyrophosphate, 2 mM sodium orthovanadate, 10 mM sodium molybdate, 5 mM sodium fluoride, 5 mM PMSF, 5 µg/mL aprotinin, and 5 ug/mL leupeptin (Sigma-Aldrich). Following collection, lysates were resolved on 10% or 14% polyacrylamide gels. Membranes were probed with antibodies against phosphorylated (p)STAT3^Tyr705^ (Cell Signaling, Danvers, MA, USA; #9145) or pSTAT1^Tyr701^ (Cell Signaling #7649), SOCS3 (SantaCruz Biotechnology, Dallas, TX, USA; #sc-73045) then stripped and probed with total STAT1 (Santa Cruz; #sc-464), total STAT3 (SantaCruz; #sc-8019) or alpha tubulin (Sigma-Aldrich;#T8203) as loading controls. Following this, the membranes were incubated with the appropriate HRP conjugated secondary antibodies (Cell Signaling; #7074 and #7076) and detection was performed using the Millipore Chemiluminescence detection kit and CL-XPosure Film (Thermo Scientific, Rockland, IL, USA) and quantified by densitometric analysis using Molecular Dynamics ImageQuant version 5.1 analysis software (Caesarea, Israel). Pixel density was normalized to α-tubulin for each sample to correct for loading differences. 

### 2.5. Methylation Analysis

Pyrosequencing assays were designed using the algorithms built into the PyroMark Assay Design Software (Version 2.0.1, Qiagen). PCR and sequencing primers for each assay are listed in Table 1. DNA samples were converted using the Epitect Bisulphite Conversion Kit (Qiagen) and PCR amplifications performed with the PyroMark PCR Kit (Qiagen). The assay was performed on PyroMark 24 Pyrosequencing System (Qiagen) and data were analyzed using PyroMark Q24 software. Methylation analysis of *SOCS1* was also performed using Illumina Infinium Human Methylation 450 BeadChip microarray (San Diego, CA, USA), as previously described [23].

### 2.6. Modulation of miRNA155 In Vitro 

Human lung fibroblasts were grown to 90% confluence then transfected with miR155 pre-cursor, miRNA precursor control, or miR155 inhibitor (Ambion Pre-miR, Life Technologies, Carlsbad, CA. USA). Cells were then treated with 20 ng/mL IFNγ and incubated for 2 h (for RNA analysis) or 2 h 45 min or 3 h for Western blot analysis of pSTAT1, SMAD2 (Santa Cruz; #sc-101153), and α-tubulin.

### 2.7. Statistical Analysis

Statistical analysis was performed in Graph Pad Prism (Graphpad Software Inc., San Diego, CA, USA) for HLF vs. IPF mRNA expression analysis a Mann–Whitney test was performed. For multiple comparisons, an Anova with a Tukey’s post-hoc test. A *p* < 0.05 deemed to be significant.

## 3. Results

### 3.1. SOCS1 mRNA Levels Are Reduced in IPF Compared with Control Lung Fibroblasts

The relative expression of *SOCS1* mRNA, detected using real time PCR, was reduced in cultured IPF fibroblasts compared to control human lung fibroblasts (HLF; Figure 1A), supporting existing studies in IPF lungs. There was a trend towards reduced *SOCS3* mRNA levels in IPF fibroblasts compared to HLF, but the difference was not significant. Although *SOCS1* mRNA levels were significantly reduced, the kinetics of IFNγ-mediated induction of *SOCS1* mRNA and activation of STAT1, measured as STAT1 phosphorylation, were not altered (Figure 1B,C). There was also a trend towards reduced SOCS3 protein expression and consistent with *SOCS1* mRNA, and there was no change in the kinetics of *SOCS3* mRNA expression in response to IL-6 (Figure 1D). However, an analysis of multiple cell cultures from different donors demonstrated that the magnitude of *SOCS3* expression varied between cultures, as shown in Figure 1D. 

### 3.2. SOCS1 Methylation in IPF

Epigenetic down-regulation of SOCS1 has been associated with altered STAT3 activation in HCC [13] and previous studies suggest that methylation is responsible for *SOCS1* gene silencing [13,24,25,26,27,28,29]. Using a bioinformatics approach [30], five putative STAT1 and STAT3 binding sites were identified within the *SOCS1* gene promoter and primers designed against these specific regions (Table 1). These STAT1/STAT3 transcription factor binding sequences (TFBS) fall within an annotated CpG Island. A group analysis performed on nine IPF and six control lung fibroblast cultures showed that there was no significant difference between the level of methylation of the *SOCS1* promoter at STAT1 and STAT3 sites (Figure 2). Less than 10% methylation was detected at the STAT3 sites and less than 5% methylation at the STAT1 sites. This lack of methylation was also confirmed in gDNA extracted from FFPE IPF (*n* = 8) and non-IPF (*n* = 4) lung tissue samples (Figure 2). Furthermore, independent analysis performed on a cohort of IPF and non-IPF patients using the Illumina Infinium Human Methylation 450 BeadChip microarray further validated our observations, showing that the *SOCS1* gene is not constitutively methylated, and also demonstrated that there is no significant change in *SOCS1* gene methylation in IPF compared to HLF (summarized in Table 2).

### 3.3. miR155 as a Potential Regulator of SOCS1 Expression in IPF Fibroblasts

miR155 has been shown to negatively regulate *SOCS1* expression [13,26,27,28,29,30]. Therefore, we measured the relative expression of miR155 in FFPE human lung biopsy tissue and showed that it was significantly increased in IPF lung samples compared to control lung tissue samples (Figure 3A, *p* = 0.0485). However, analysis of miR155 in fibroblasts provided the opposite result; there was reduced expression in IPF fibroblasts compared with HLF (Figure 3B, *p* = 0.0028) despite IPF fibroblasts having reduced *SOCS1* mRNA levels. 

We next sought to determine whether *SOCS1* was a target of miR-155 by overexpressing the miR-155 in control human lung fibroblasts (CCD-19Lu). Cells were transfected with either a miR155 precursor or control precursor and basal or IFNγ-induced *SOCS1* levels were determined by real time PCR. Despite a significant increase in miR155 levels (Figure 3C), no change in basal or IFNγ-induced *SOCS1* mRNA levels were detected in pre-miR155-transfected fibroblasts (Figure 3D). This suggests that the regulatory effects of miR155 on *SOCS1*, if any, are at the level of translation in these cells [31]. The level of IFNγ-induced STAT1 phosphorylation was examined in miR155-transfected lung fibroblasts. Phospho-STAT1 expression was reduced in miR155 over-expressing cells and not increased as predicted (Figure 3E). The activity of miR155 in cells transfected with pre-miR155 was confirmed by Western blot analysis of SMAD2, a known target of miR155. SMAD2 protein levels were reduced in pre-miRNA-transfected fibroblasts compared with control (Figure 3E). 

We next examined the role of miR155 overexpression on transforming growth factor beta (TGFβ)-induced collagen production using the scar-in-a-jar assay, an in vitro molecular crowding assay of collagen deposition [21]. Primary lung fibroblasts were cultured in the presence of ficoll 70, ficoll 400, and L-ascorbate, and collagen deposition was quantified. Significant collagen deposition by primary human control lung fibroblasts was observed in cultures treated with 1 ng/mL of TGFβ (Figure 4A). Following transfection, a significant increase in miR155 was detected in cells transfected with pre-miR155 compared to untreated, those transfected with CTR pre-miR or miR155 inhibitor (Figure 4B). Analysis of TGFβ-induced collagen production revealed no significant difference in TGFβ-induced collagen deposition in pre-miR155-treated cells compared to control cultures treated with CTR pre-miR or miR155 inhibitor (Figure 4C).

## 4. Discussion

Exaggerated and prolonged STAT3 activation is a feature of many fibrotic disorders including liver, kidney, and lung fibrosis [4,5,6], as well as a number of cancers [32,33,34]. In IPF lungs and those of bleomycin treated mice, STAT3 activation is localized to cells adjacent to fibrotic foci [4,5]. The mechanism responsible for elevated STAT3 levels at sites of interstitial fibrosis is not fully understood, but studies in liver [13] and kidney [35] suggest that they may be due to reduced SOCS1 expression and activity. Based on the hypothesis that exaggerated STAT3 activity in IPF is due to altered expression and activity of SOCS proteins, this study investigated mechanisms that may contribute to reduced SOCS1 expression in IPF fibroblasts. 

Reduced expression of SOCS1 in IPF lung tissue and fibroblasts cultured from the lungs of IPF patients was confirmed in this study. *SOCS1* and *SOCS3* mRNA kinetics in HLF and IPF fibroblast cultures was examined by treating cells with IFNγ and IL-6, respectively, over time. The kinetics of IFNγ-induced *SOCS1* and IL-6-induced *SOCS3* were not altered in either HLF or IPF fibroblasts, suggesting that the induction of mRNA was not significantly altered in these cells. 

Hyper-methylation of the *SOCS1* promoter has been associated with reduced SOCS1 levels in hematopoetic malignances, gastric cancer [36], and hepatocellular carcinoma [13], and therefore the possibility that reduced SOCS1 levels in IPF lungs was due to increased methylation of the SOCS1 promoter was investigated. In addition to STAT1, putative STAT3 binding sites were identified within the promoter region of *SOCS1*. Neither the level of methylation of the *SOCS1* promoter at these putative STAT transcription factor binding sites or across the *SOCS1* gene was different between IPF and HLF fibroblasts, and IPF and non-IPF lung tissue, suggesting that this is not the mechanism of regulation of *SOCS1* expression in IPF lung tissue generally and fibroblasts in particular.

A number of recent studies have investigated miRs in the context of lung fibrosis, with let7, miR21, miR29, and miR155 all differentially expressed in IPF (reviewed by Pandit et al. [37]). Of these, miR155 has been shown to target and negatively regulate *SOCS1* [26,27,30,38,39]. Indeed, tumor cell invasiveness and metastasis is regulated by the effect of miR155 on the STAT3-SOCS1 pathway [26,27,38,39] and SOCS1 function is reduced in human breast cancer cells expressing high levels of miR155 [27]. The effects of miR155 have also been described as pro-fibrotic [37,40]. Knockdown of miR155 significantly blocked myofibroblast differentiation of rat kidney fibroblasts in vitro and reduced renal fibrosis in a rat model [30]. Up-regulation of miR155 expression also correlated with enhanced bleomycin-induced fibrosis in the mouse [40] and the extent of lung damage observed in patients with IPF [41]. In this study, an increase in miR155 expression in human lung tissue from patients with IPF was observed, although the levels were lower in IPF fibroblasts. This suggests that increased miR155 in IPF tissue is not fibroblast-related. Increasing the levels of miR155 also had no effect on SOCS1 expression in HLF or IPF fibroblasts, questioning the role of miR155 in regulating SOCS1 in fibroblasts. miR155 may still regulate *SOCS1* in other lung cells such as epithelial cells, which express high levels of pSTAT3 in IPF lung tissue [42] or may regulate other pro-fibrotic mediators. 

SOCS1 is not the only target of miR155. An online database search using miRTarBase revealed more than 245 targets for miR155. Angiotensin II type I receptor (AT1R) has also been reported as a target for miR155 activity. AT1R activity has been shown to be required for bleomycin-induced collagen production [42,43]. Furthermore, TGFβ signaling molecules are also targeted by miR155. Indeed, in this study, transfection of miR155 precursor molecules into fibroblasts resulted in down regulation of SMAD2. This suggests that miR155 could influence fibrosis by regulating TGFβ-induced collagen production. However, using a molecular crowding assay (scar-in-the-jar assay), we did not see any difference on TGFβ-induced collagen production following transfection of miR155 precursor molecules. This may reflect the capacity of other Smad molecules to compensate for the loss of SMAD2. However, more detailed analysis of miR155/Smad regulation and its effect on ECM deposition needs to be performed and will be the subject of future studies. 

Contrary to these findings, Kurowska-Stolarska and colleagues showed that mice depleted in miR155 (miR155^−/−^) actually had exacerbated lung fibrosis following bleomycin treatment, and suggested that this was due to deregulation of the miR-155 target gene the liver X receptor (LXR)α in lung fibroblasts and macrophages [44]. Inhibiting LXRα in bleomycin treated miR155^−/−^ mice reduced the exacerbated fibrotic response. They also showed that IPF fibroblasts had increased levels of LXRα and they could reduce the fibrotic phenotype of IPF and miR-155^−/−^ fibroblasts by transfecting the cells with miR155. In our study, we only looked at the effect of overexpressing miR155 on TGF-β-induced collagen production but saw no change. However, we did not look in the absence of TGF-β. Clearly, the role of miR155 in lung fibrosis and IPF is complicated, possibly due to variable expression in different cells and at different stages of disease and the levels of its various targets. However, despite these mixed findings, miR155 still presents as an interesting molecule to follow up, as the levels of miR155 have been shown to be elevated in the serum [41] as well as in lung tissues (Figure 2) from IPF patients. miR155 may also represent a biomarker that can be used to assess disease progression, but a larger, longitudinal study would be required to confirm this. 

In conclusion, we confirmed reduced *SOCS1* mRNA levels in lung tissue and cultured fibroblasts from IPF patients compared with controls, but this change was not due to changes in the kinetics of induction of *SOCS1* mRNA, DNA methylation, or miR155. We were also unable to find a significant role for SOCS1 in regulating lung fibroblast function, but SOCS1 may regulate other cells in the lung that are important in the fibrotic process.

## Figures and Tables

**Figure 1 biomedicines-09-00498-f001:**
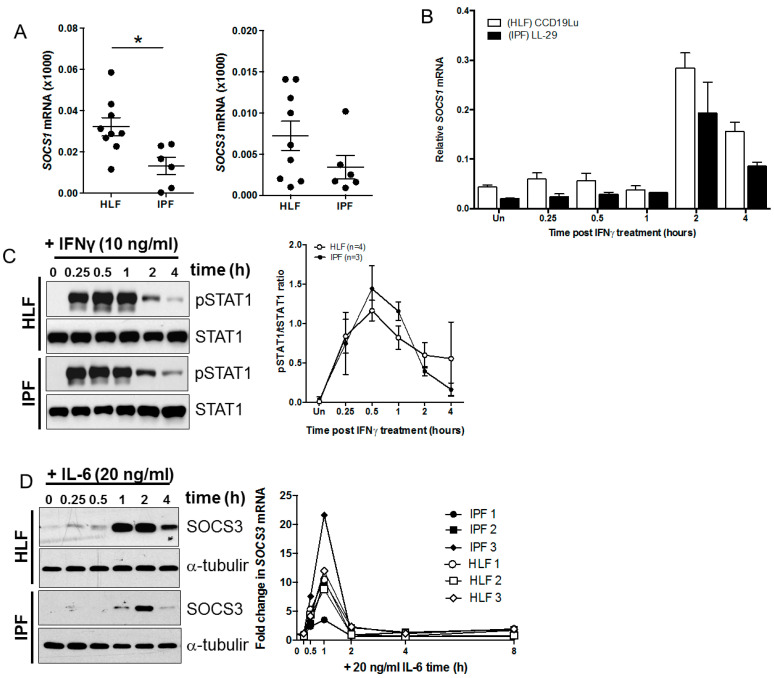
(**A**) SOCS1 is down regulated in IPF fibroblasts. *SOCS1* and *SOCS3* mRNA levels were determined in IPF (*n* = 6) and control lung fibroblasts (HLF, *n* = 9) by qPCR and gene expression normalized to the housekeeping gene 18S. Data are represented as relative *SOCS* mRNA expression. *SOCS1* levels were significantly reduced in IPF fibroblasts (*p* = 0.0120). A trend towards reduced *SOCS3* levels in IPF fibroblasts was also detected (*p* = 0.1532). (**B**) The capacity for IFNγ to induce *SOCS1* 2 and 4 h following treatment with 10 ng/mL IFNγ was not altered. (**C**) Similarly, the kinetics of IFNγ-induced pSTAT1 and was not altered in in IPF fibroblasts (HLF *n* = 4, IPF *n* = 3 cell cultures for pSTAT1). (**D**) IL-6-induced SOCS3 mRNA and protein were not altered in IPF fibroblasts, although the magnitude of the response was varied (HLF and IPF *n* = 3 for *SOCS3* mRNA kinetics).

**Figure 2 biomedicines-09-00498-f002:**
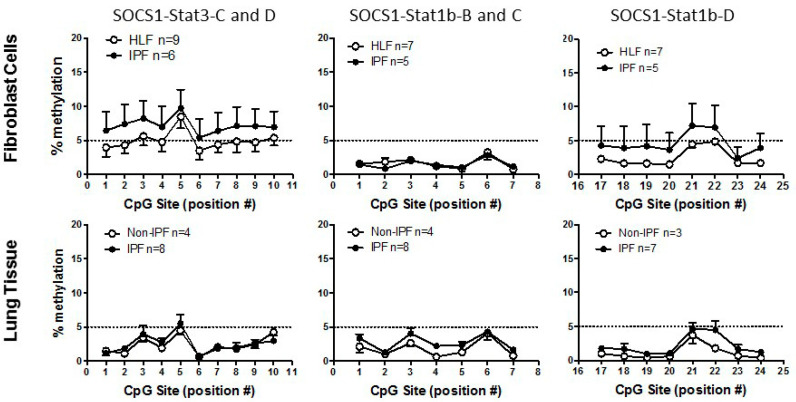
Approximately 500 bp of sequence surrounding the stat1 and stat3 transcription factor binding sites (TFBS) in the promoter region of *SOCS1* were input into the software. CpG sites within or adjacent to the predicted TFBS were selected as target sites for analysis and primers designed to target these sites were chosen from a list generated by the software on the basis of the algorithms predicted assay quality. PCR sequencing primers for each assay are listed in Table 1. DNA samples were converted using the Epitect Bisulphite Conversion Kit (Qiagen). All PCR amplifications were performed with the PyroMark PCR Kit (Qiagen) and products visualized on a 2% agarose gel to confirm quality. The amplicons containing the CpG target sites were PCR amplified using a biotin labeled, HPLC purified forward primer and standard sequencing grade reverse primer. The PCR product was bound to streptavidin sepharose high performance beads (GE Healthcare Life Sciences), following a washing and denaturation step the beads were transferred to optically clear, 24 well sequencing plate in the presence of the pyrosequencing primer. Pyrosequencing was performed on a PyroMark 24 Pyrosequencing System (Qiagen) as per the manufacturer’s instructions. Data were analyzed on the PyroMark Q24 software to give the % methylation values for each CpG site in the sample. In this study, three assays were performed, SOCS1-Stat3-C and D; SOCS1-Stat3b-B and C and SOCS1-Stat1-D (Table 1). Analysis of control human lung fibroblasts (HLF) *n* ≥ 7 or IPF fibroblasts *n* ≥ 5 are graphed. Similar analysis was performed on DNA extracted from formalin-fixed paraffin-embedded lung tissue samples with non-IPF (*n* ≥ 3) and IPF (*n* ≥ 7). Methylation levels were low in all samples, with less than 5% methylation detected in all but SOCS1-Stat3C-D assay of HLF vs. IPF. These data suggest that methylation of putative STAT1 and STAT3 TFBS within the *SOCS1* promoter are not altered in IPF.

**Figure 3 biomedicines-09-00498-f003:**
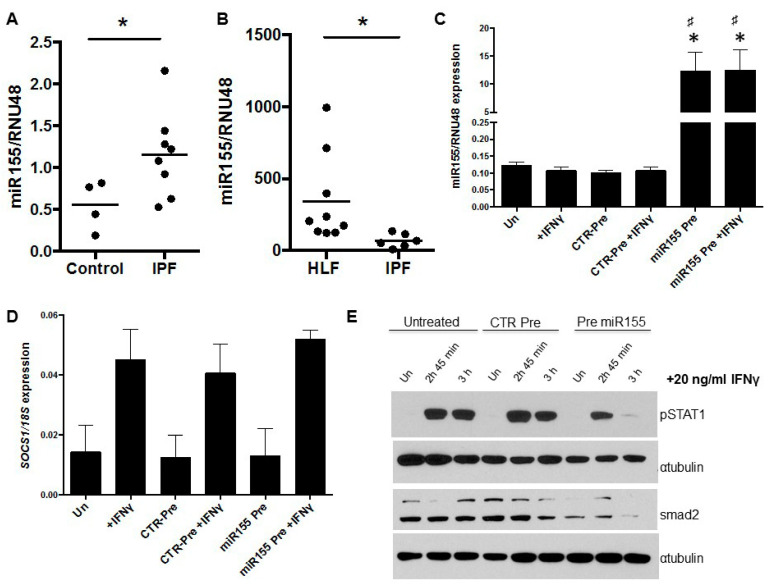
miR155 is altered in lung fibroblasts but does not regulate SOCS1 function. miR155 levels were measured in cDNA generated from (**A**) FFPE lung tissue extracts and (**B**) lung fibroblasts cultures. miR155 levels (normalized to RNU48) are elevated in the lungs of patients with IPF (*n* = 8) compared to control lung tissue (*n* = 4) *p* = 0.0485 but are significantly reduced in IPF fibroblasts (*n* = 6) compared to control lung fibroblasts (*n* = 9), *p* = 0.0028. (**C**) miR155 levels were increased in cultures of CCD-19Lu fibroblasts using a precursor to miR155 (pre-miR155). * *p* < 0.05 compared to un-transfected controls, # *p* < 0.05 compared to CTR pre-miR transfected cells, *n* ≥ 3). (**D**) High levels of miR155 did not alter basal levels of *SOCS1* mRNA or the capacity for IFNγ to induce *SOCS1* mRNA. (**E**) Western blot analysis revealed reduced phosphorylated STAT1 and SMAD2 levels in cells transfected with the pre-miR155. Alpha-tubulin was used as a control for protein loading.

**Figure 4 biomedicines-09-00498-f004:**
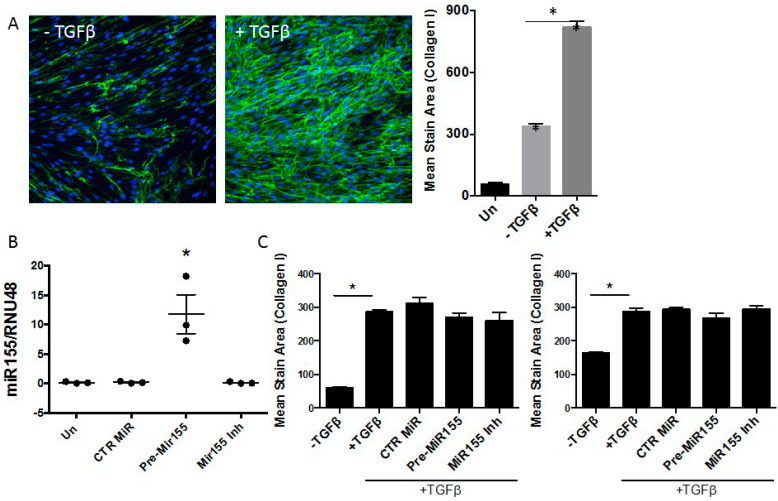
miR155 overexpression does not alter TGFβ-induced collagen deposition in vitro. (**A**) Using the in vitro scar-in-a-jar model, enhanced collagen deposition was detected by primary human lung fibroblasts cultured with 1 ng/mL of TGFβ in the presence of ficoll and L-ascorbate for 48 h. Collagen deposition was quantified at ×20 magnification using ImageXpress and is represented as mean stain area ± SEM, *n* ≥ 5. (**B**) miR155 levels were significantly increased in pre-miR155-transfected lung fibroblasts. PCR data from three individual primary lung fibroblast cultures are shown. (**C**) Quantitative analysis of collagen deposition in two individual primary lung fibroblast cultures transfected with either control precursor miR (CTR MiR), precursor to miR155 (pre-miR155) or a miR155 inhibitor (miR155 Inh). No significant difference was detected in pre-miR155 transfected or TGFβ only groups, suggesting that miR155 overexpression does not affect TGFβ-induced collagen deposition in this model. * *p* < 0.05.

**Table 1 biomedicines-09-00498-t001:** Primer sequences for pyrosequencing assays.

Assay Name	Primer	Primer Sequence
**SOCS1-Stat1b-B and C**	Forward PCR primer	GAGGGTTTAGAAGAGAGGGAAATAG
	Reverse PCR Primer	5’Biotin-CCCCCAACTCCACTTTTATTT
	Sequencing primer	GAAGAGAGGGAAATAGG
**SOCS1-Stat1b-D**	Forward PCR primer	GGTTTGATTATAGGTTTTAGAGGAATTT
	Reverse PCR Primer	5’Biotin-CCCCAACCTCAATTTCTC
	Sequencing primer	ATTATAGGTTTTAGAGGAATTTT
**SOCS1-Stat3-C and D**	Forward PCR primer	GGGGTTTTTTTGAAGTTTGTGGTTAG
	Reverse PCR Primer	5’Biotin-CCCCTCCTAACCCCTACTC
	Sequencing primer	AGTTTGTGGTTAGGT

**Table 2 biomedicines-09-00498-t002:** Methylation analysis of the *SOCS1* gene was performed on six control lung fibroblast cultures and five IPF lung fibroblast cultures using the Illumina Infinium Human Methylation 450 BeadChip microarray. Both basal levels of methylation (untreated (UT)) and the level of methylation in 5′-aza-2′-deoxycytidine (5′aza)-treated (T) cells were measured. Cell cultures were treated daily with 1 μM of 5′-aza for more than a week to ensure at least three population doublings as previously described [22]. For microarray data, an adjusted *P* value of 0.05 and a Δβ value ≥ 0.136 was considered significant [25,26,27,28,29,30,31]. Data are represented as % methylation at a given CpG site within the *SOCS1* gene. No significant change in the level of methylation was detected at any of the sites tested or in the four comparisons tested; CTR UT vs. CTR T, IPF UT vs. IPF T, CTR UT vs. IPF UT, CTR T vs. IPF T (*n* = 6 CTR and *n* = 5 IPF) fibroblast cell cultures.

CpG ID	% Methylation								Change % Methylation							
	CTR UT (*n* = 6)		CTR T (*n* = 6)		IPF UT (*n* = 5)		IPF T (*n* = 5)		CTR UT vs. CTR T		IPF UT vs. IPF T		CTR UT vs. IPF UT		CTR T vs. IPF T	
	Mean	SD	Mean	SD	Mean	SD	Mean	SD	Change	*p* Value	Change	*p* Value	Change	*p* Value	Change	*p* Value
cg00487159	11.713	1.195	14.184	2.885	11.790	1.135	14.947	4.414	2.472	0.811	3.157	0.863	0.077	0.983	0.762	0.969
cg00674265	15.652	2.105	18.261	3.754	17.276	1.673	19.941	3.104	2.609	0.891	2.665	0.855	1.624	0.669	1.680	0.917
cg01330880	5.561	1.180	5.060	1.770	5.159	0.956	4.460	1.141	−0.501	1.000	−0.699	1.000	−0.402	0.887	−0.600	0.948
cg01717706	9.709	1.131	10.844	2.073	10.423	1.493	12.551	2.698	1.135	1.000	2.128	0.874	0.714	0.810	1.707	0.876
cg03014241	32.513	13.765	31.706	11.822	19.027	5.034	23.322	6.284	−0.807	1.000	4.295	0.913	−13.486	0.469	−8.384	0.816
cg03195600	6.103	1.499	5.644	1.178	5.216	1.090	7.457	2.502	−0.459	1.000	2.242	0.853	−0.888	0.761	1.813	0.828
cg03553358	14.261	3.334	17.253	5.228	17.211	4.445	18.966	6.256	2.992	0.979	1.755	1.000	2.950	0.684	1.713	0.952
cg04004558	22.836	4.740	24.374	4.471	20.942	6.490	25.944	4.320	1.538	1.000	5.002	0.871	−1.894	0.872	1.570	0.942
cg04266460	2.923	0.626	3.668	1.114	3.271	0.444	3.116	0.981	0.746	1.000	−0.155	1.000	0.348	0.858	−0.553	0.940
cg04289163	1.890	0.404	1.648	0.257	1.711	0.513	1.867	0.122	−0.242	1.000	0.156	1.000	−0.179	0.931	0.219	0.972
cg04609482	11.691	1.423	13.629	3.087	11.725	1.424	13.837	2.642	1.938	0.926	2.112	0.871	0.034	0.994	0.208	0.991
cg05181221	2.208	0.422	2.249	0.288	2.102	0.436	2.144	0.375	0.041	1.000	0.041	1.000	−0.106	0.961	−0.106	0.988
cg05701125	3.816	1.301	3.615	0.475	7.013	2.437	6.152	2.259	−0.201	1.000	−0.861	1.000	3.197	0.367	2.537	0.648
cg05730996	8.417	1.043	9.603	1.854	8.248	0.785	10.522	2.465	1.186	0.972	2.274	0.852	−0.169	0.954	0.919	0.935
cg05766667	6.259	0.312	5.939	1.122	4.897	1.079	4.925	1.198	−0.320	1.000	0.028	1.000	−1.362	0.464	−1.014	0.883
cg05902273	7.706	0.840	9.699	2.405	7.593	1.150	8.733	1.512	1.993	0.821	1.140	0.941	−0.113	0.971	−0.966	0.929
cg06082432	6.052	1.483	5.389	1.249	6.880	0.831	5.944	1.423	−0.663	1.000	−0.936	0.972	0.828	0.765	0.556	0.948
cg06220235	4.380	1.626	5.034	3.023	4.748	1.054	4.409	0.495	0.653	1.000	−0.339	1.000	0.368	0.918	−0.624	0.961
cg06295404	3.795	0.746	3.422	0.912	3.135	0.501	3.817	1.184	−0.374	1.000	0.682	1.000	−0.660	0.726	0.396	0.959
cg09567644	6.622	1.087	5.526	1.012	5.201	1.202	4.890	0.731	−1.097	0.902	−0.311	1.000	−1.421	0.556	−0.636	0.923
cg10513253	1.958	0.331	1.875	0.411	1.897	0.349	1.727	0.498	−0.083	1.000	−0.170	1.000	−0.061	0.978	−0.148	0.983
cg10784813	26.767	14.114	27.686	12.717	21.366	7.658	23.734	6.576	0.919	1.000	2.368	1.000	−5.401	0.815	−3.952	0.935
cg11973052	1.450	0.288	1.675	0.309	1.181	0.198	1.321	0.225	0.225	1.000	0.140	1.000	−0.269	0.874	−0.354	0.951
cg12642006	19.242	3.395	21.525	3.090	24.291	3.112	28.573	4.704	2.283	0.973	4.283	0.852	5.049	0.384	7.049	0.523
cg26012103	7.832	1.312	8.621	1.213	8.350	1.031	9.473	2.089	0.789	1.000	1.123	0.981	0.518	0.859	0.852	0.926
cg26301908	8.050	1.290	8.205	0.863	9.517	1.597	9.057	1.929	0.155	1.000	−0.460	1.000	1.467	0.616	0.852	0.917
cg27003951	11.410	2.728	14.376	2.487	11.229	0.894	16.208	5.881	2.966	0.805	4.979	0.852	−0.181	0.975	1.833	0.929
cg27540841	3.287	0.640	2.876	0.320	3.181	0.419	3.256	0.573	−0.410	1.000	0.075	1.000	−0.105	0.964	0.380	0.950

## Data Availability

Data is contained within the article.

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
