# Peer review of "Reduced SOCS1 Expression in Lung Fibroblasts from Patients with IPF Is Not Mediated by Promoter Methylation or Mir155"

_biomedicines, 2021, doi:10.3390/biomedicines9050498_

Round 1

Reviewer 1 Report

The manuscript presents some interesting findings. They confirmed that in IPF, SOCS1 expression is decreased. Then the authors investigated whether DNA methylation in the promoter region of SCOS1 or mir-155 are involved in the diminished expression of SOCS1 and they concluded neither was.

I do have a few comments.

  1. What happened to the figure legends?
  2.  How are Western blots quantified?
  3. Since methylation is a main part of the story, I'd like to see the S1 figure moved to the main part.
  4. Figure 2E. What was the rationale for choosing two time points at 2hr:45min and 3 hrs? Why is there such big variation of loading controls (alpha-tubulin) among different groups?
  5. Figure 3. Where is the scale bar?
  6. Figure 3. In 3C, what is the difference between the two panels? This needs to be described in results and figure legends.
  7. The following paper should be cited.  Expression of suppressor of cytokine signaling 1 in the peripheral blood of patients with idiopathic pulmonary fibrosis. Bao Zhiyao et al. 
  8. The role of mir-155 is controversial at best. Please see " The role of microRNA-155/liver X receptor pathway in experimental and idiopathic pulmonary fibrosis" by Mariola Kurowska-Stolarska et al for a different opinion.

Author Response

We thank the reviewer for the comments and suggestions. We have addressed each comment below.

  1. What happened to the figure legends?

We apologise for this error. Unfortunately, the manuscript was not uploaded correctly, and the figure legends did not get added. To make sure this doesn’t happen again, the manuscript has been uploaded in the standard manner rather than in published format.  

  1. How are Western blots quantified?

Western blots were quantified by densitometric analysis which has now been included in the methods section.

  1. Since methylation is a main part of the story, I'd like to see the S1 figure moved to the main part.

Figure S1 has now been moved to the main text as figure 2. Old figure 2 and 3 are now figures 3 and 4 respectively. Supplementary table 1 has also been moved into the main text of the manuscript.

  1. Figure 2E. What was the rationale for choosing two time points at 2hr:45min and 3 hrs? Why is there such big variation of loading controls (alpha-tubulin) among different groups?

Initial experiments showed that these were the optimum time points to show maximum protein expression. The analysis was performed on the same sample set but were run on two different gels for Stat3 and Smad2 explaining the variability with the loading controls.

  1. Figure 3. Where is the scale bar?

Images were taken at x20 magnification (ImageXpress) which is included in the figure legend. Scale bars were not available. 

  1. Figure 3. In 3C, what is the difference between the two panels? This needs to be described in results and figure legends.

These represent two separate cultures with replicates within the expt. This is stated in the figure legend.  

  1. The following paper should be cited.  Expression of suppressor of cytokine signaling 1 in the peripheral blood of patients with idiopathic pulmonary fibrosis. Bao Zhiyao et al. 

This paper has now been cited in the introduction.

  1. The role of mir-155 is controversial at best. Please see " The role of microRNA-155/liver X receptor pathway in experimental and idiopathic pulmonary fibrosis" by Mariola Kurowska-Stolarska et al for a different opinion.

We thank the reviewer for pointing this paper out to us. We have now discussed this paper in the discussion section.

Reviewer 2 Report

The work entitled “Reduced SOCS1 expression in lung fibroblasts from patients  with IPF is not mediated by promoter methylation or miR155” submitted to Biomedicines requires major corrections before further processing. Please find some comments that should be considered in the revision of your manuscript:

Major points:

  1. Please avoid comment on results and conclusion in the introduction part, please focus on the aims.
  2. The description of cell culture is unclear and requires careful improvement.
  3. Please briefly describe methods of fibroblasts isolation and identification.
  4. Who were the cells isolated from? Please provide a brief description of the patients, the consent of the appropriate ethical committee, what group were the control patients, how were they qualified to the control group? Please clearly indicate which tests were performed on isolated cells and which on commercial ones
  5. Please enrich M&M part: specify antibodies used in the study, add catalog numbers, what buffer was used for protein isolation, how much protein was applied for electrophoresis? how much of RNA and cDNA was used in the qPCR experiments ? please add ATCC numbers for commercial cells, provide Taqman probes numbers, how the cells were quiesced, transected etc.

Results

  1. Key note: The text is missing figure captions and it is impossible to correctly read the results in this form.
  2. Please justify in the text the use of IL-6 and IFN-gamma in the research.
  3. Fig. 1B,C,D, 2D - what about the statistical significance of the results presented in these graphs
  4. Fig. 2E - Please explain why smad2 was tested, what do the two bands in the image mean, was the phosphorylation of this protein checked?
  5. I suggest that you consider adding the methylation results to the body of the manuscript

Discussion

  1. The whole discussion part needs improvement, please avoid duplicating the information contained in the introduction, focus on your own results. A few sentences and leading ideas are raised several times in the discussion - please avoid it. In the summary, please include the most important conclusions from the obtained results.

Minor points:

  1. Please check references and pages numbering (e.g. in the M&M part some references possess number 46-48).
  2. There are several typos in both the text and the figures, e.g. in the names of proteins.

Author Response

We thank the reviewer for the comments and suggestions. We have addressed each comment below.

  1. Please avoid comment on results and conclusion in the introduction part, please focus on the aims.

Different journals have different approaches when it comes to including a summary of results and conclusions in the introduction. Several other published manuscripts in Biomedicines include this information in the introduction and so I will leave it in at this stage unless the editor specifically tells me to remove it. However, I have more clearly stated the aims of the study in the introduction.

  1. The description of cell culture is unclear and requires careful improvement.

We have included more information in the methods section but to keep the number of words within the required limits, a lot of the specific details are included in the figure legends. Unfortunately, due to loading errors, the figure legends were not included in the original submitted manuscript.

  1. Please briefly describe methods of fibroblasts isolation and identification.

Primary cell cultures either came from ATCC or were generated by our collaborators at UCL using the protocol described in (Keerthisingam et al. 2001 - cited in the manuscript). This has been clarified in the methods.

  1. Who were the cells isolated from? Please provide a brief description of the patients, the consent of the appropriate ethical committee, what group were the control patients, how were they qualified to the control group? Please clearly indicate which tests were performed on isolated cells and which on commercial ones.

IPF fibroblasts were isolated from fibrotic lung tissue from patients undergoing surgical lung biopsy or transplant surgery (aged 62 ± 4 years). Control cells were isolated from lung tissue obtained from histologically normal areas of peripheral lung removed at lung cancer resection (aged 59 ± 7 years). All tissue was obtained with appropriate informed consent and its use approved by the East Midlands – Nottingham 2 NRES Committee, Ref. 12/EM/0058. All experiments were conducted in accord with the terms of the informed consents and in accordance with relevant guidelines and regulations. Tests were performed on all IPF and control cell cultures from London ATCC. miR-155 transfection studies were performed on ATCC CCD-19Lu fibroblasts.

  1. Please enrich M&M part: specify antibodies used in the study, add catalog numbers, what buffer was used for protein isolation, how much protein was applied for electrophoresis? how much of RNA and cDNA was used in the qPCR experiments ? please add ATCC numbers for commercial cells, provide Taqman probes numbers, how the cells were quiesced, transfected etc.

Catalogue numbers have been included for all antibodies. Their inclusion can be determined by the editorial team. Protein and RNA isolation buffers etc have been added to the methods. ATCC numbers have been added for the commercial cells. Taqmqn probe numbers have been included.

  1. Key note: The text is missing figure captions and it is impossible to correctly read the results in this form.

As previously explained, this was due to an error loading the manuscript and I sincerely apologise. I fully understand how difficult it must have been trying to understand the manuscript results.

  1. Please justify in the text the use of IL-6 and IFN-gamma in the research.

The reason for using IL-6 and INF-γ as activators of STAT3 and STAT1 respectively has been stated in the methods section.

  1. 1B,C,D, 2D - what about the statistical significance of the results presented in these graphs?

The statistical significance is stated in the figure legends.

  1. 2E - Please explain why smad2 was tested, what do the two bands in the image mean, was the phosphorylation of this protein checked?

As explained in the results and discussion, Smad2 is also a target for miR155. Hence, we examined Smad2 levels after miR155 transfection to show that the transfected pre-miR155 was working. miR155 downregulates total Smad2, not phosphorylation of Smad2.

  1. I suggest that you consider adding the methylation results to the body of the manuscript

We agree and have now added the methylation data to the main body of the manuscript.

11. The whole discussion part needs improvement, please avoid duplicating the information contained in the introduction, focus on your own results. A few sentences and leading ideas are raised several times in the discussion - please avoid it. In the summary, please include the most important conclusions from the obtained results.

The discussion has been condensed and repetition removed.

12. Please check references and pages numbering (e.g. in the M&M part some references possess number 46-48).

Referencing checked and corrected.

  1. There are several typos in both the text and the figures, e.g. in the names of proteins.

Typos corrected where identified.

Round 2

Reviewer 1 Report

The authors properly addressed my concerns and the quality of the manuscript has improved significantly.